# Biallelic *TANGO1* mutations cause a novel syndromal disease due to hampered cellular collagen secretion

Caroline Lekszas[1‡], Ombretta Foresti[2‡], Ishier Raote[2‡], Daniel Liedtke[1], Eva-Maria König[1], Indrajit Nanda[1], Barbara Vona[1,3], Peter De Coster[4], Rita Cauwels[4], Vivek Malhotra[2†*], Thomas Haaf[1†*]

[1]Institute of Human Genetics, Julius Maximilians University Würzburg, Würzburg, Germany; [2]Centre for Genomic Regulation, The Barcelona Institute of Science and Technology, Barcelona, Spain; [3]Department of Otorhinolaryngology, Head and Neck Surgery, Tübingen Hearing Research Centre (THRC), Eberhard Karls University Tübingen, Tübingen, Germany; [4]Department of Pediatric Dentistry and Special Care, PaeCoMeDis Research Group, Ghent University Hospital, Ghent, Belgium

**Abstract** The transport and Golgi organization 1 (TANGO1) proteins play pivotal roles in the secretory pathway. Full length TANGO1 is a transmembrane protein localised at endoplasmic reticulum (ER) exit sites, where it binds bulky cargo within the ER lumen and recruits membranes from the ER Golgi intermediate compartment to create an exit route for their export. Here we report the first TANGO1-associated syndrome in humans. A synonymous substitution that results in exon eight skipping in most mRNA molecules, ultimately leading to a truncated TANGO1 protein was identified as disease-causing mutation. The four homozygously affected sons of a consanguineous family display severe dentinogenesis imperfecta, short stature, various skeletal abnormalities, insulin-dependent diabetes mellitus, sensorineural hearing loss, and mild intellectual disability. Functional studies in HeLa and U2OS cells revealed that the corresponding truncated TANGO1 protein is dispersed in the ER and its expression in cells with intact endogenous TANGO1 impairs cellular collagen I secretion.

**\*For correspondence:**
vivek.malhotra@crg.eu (VM);
thomas.haaf@uni-wuerzburg.de
(TH)

[†]These authors contributed
equally to this work
[‡]These authors also contributed
equally to this work

**Competing interest:** See
page 14

**Reviewing editor:** Reinhard
Fässler, Max Planck Institute of
Biochemistry, Germany

## Introduction

Collagens are the most abundantly secreted molecules in mammals and needed throughout the whole body for bone mineralization, skin and tissue assembly. Within the endoplasmic reticulum (ER) lumen, newly synthesized procollagens assemble into rigid, rod-like triple helices that are too large for export by the conventional coat protein complex II (COPII)-coated vesicle size of 60–90 nm (*Malhotra and Erlmann, 2015*; *Miller and Schekman, 2013*). Over the past years, the ER exit site (ERES) located, transport and Golgi organization one protein (TANGO1) has been identified as a player in the export of bulky cargoes like the collagens (*Bard et al., 2006*; *Malhotra and Erlmann, 2011*; *Malhotra and Erlmann, 2015*; *Raote and Malhotra, 2019*; *Raote et al., 2018*; *Raote et al., 2017*; *Saito et al., 2009*; *Santos et al., 2016*; *Wilson et al., 2011*). *TANGO1* is conserved throughout most metazoans and ubiquitously expressed in humans. It comprises 8,142 bp located at chromosome 1q41 and encodes two distinct isoforms, full length TANGO1 and TANGO1-short. Full length TANGO1 consists of 1907 amino acids (aa) and contains an N-terminal signal sequence followed by a Src-homology 3 (SH3)-like domain and a coiled-coil domain in the lumenal portion, as well as two additional coiled-coil domains (CC1 and CC2) and a proline-rich domain (PRD) in the cytoplasmic portion (*Figure 1A*). TANGO1-short is composed of 785 aa and lacks the lumenal portion contained in TANGO1 (*Saito et al., 2009*). Together with cTAGE5 encoded by the TANGO1-

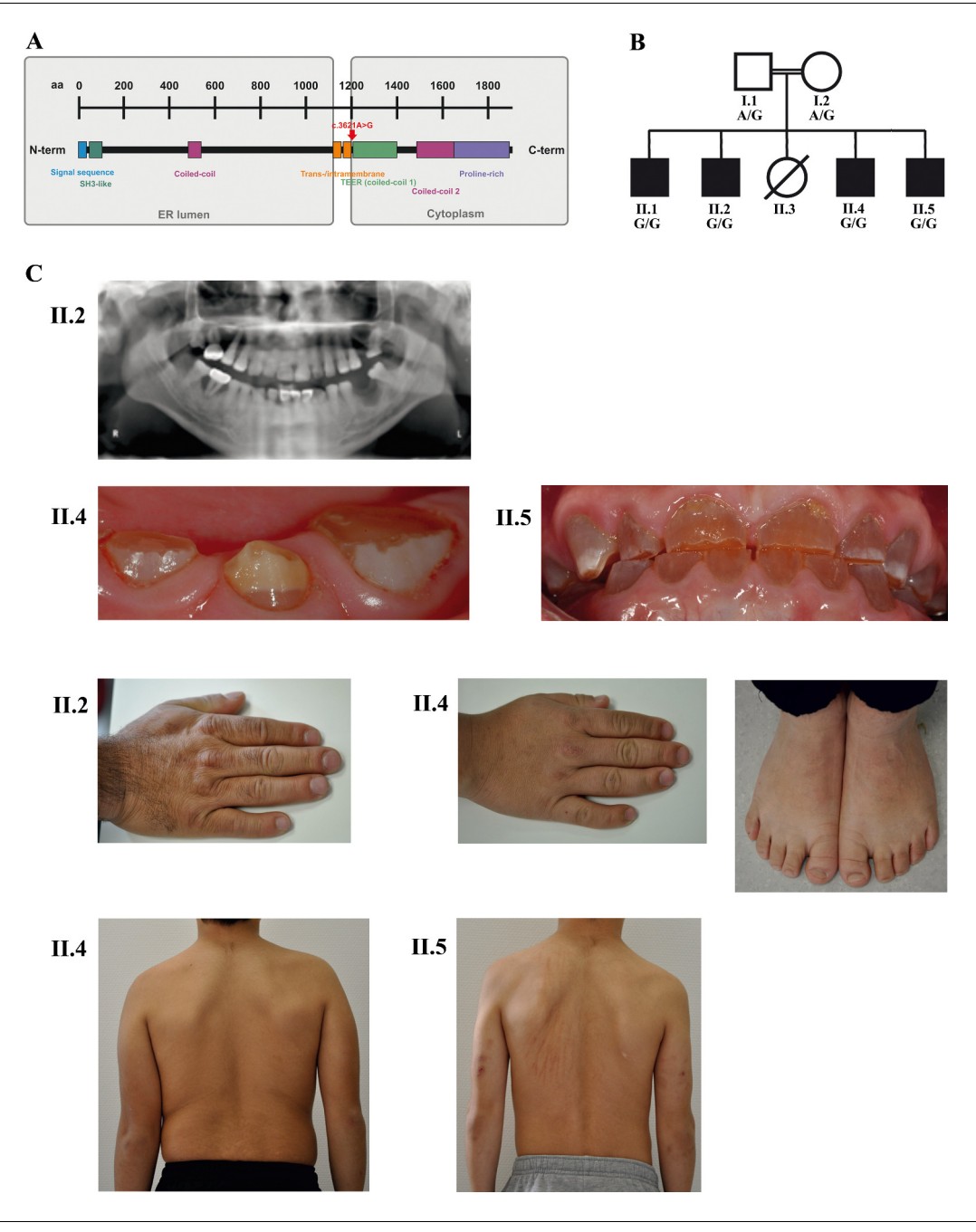

**Figure 1.** A novel syndrome caused by biallelic *TANGO1* mutations in a consanguineous family. (**A**) Structure of TANGO1 protein. The lumenal portion contains an N-terminal signal sequence followed by an SH3-like domain required for cargo binding, as well as a coiled-coil domain. A trans- and intramembrane domain anchors TANGO1 within the ER membrane. The cytoplasmic portion consists of two coiled-coil domains (CC1, also named TEER, and CC2) and a proline-rich domain at the C-terminus. The identified mutation affects residue 1207 (p.(Arg1207=)) between the intramembrane and the CC1 domain at the beginning of the cytoplasmic portion. (**B**) Pedigree of the studied family. Filled or clear symbols represent affected or unaffected individuals, respectively. The parents (I.1 and I.2) are first cousins. The four affected sons (II.1, II.2, II.4, and II.5) share a homozygous *TANGO1* (c.3621A > G) variant. The healthy child II.3 died in a household accident at the age of 16. (**C**) Dental and skeletal abnormalities of the affected brothers II.2, II.4, and II.5. Note the brachydactyly of hands and feet, clinodactyly of the fifth finger, dentinogenesis imperfecta (including an opalescent tooth discoloration with severe attrition affecting the primary and permanent dentition, as well as juvenile periodontitis, bulbous crowns, long and tapered roots, and

*Figure 1 continued on next page*

*Figure 1 continued*

obliteration of the pulp chamber and canals in the permanent dentition), the skin lesions due to pruritus in all affected children; and the scoliosis in II.4 and II.5.

---

like protein gene (*TALI*), TANGO1 and TANGO1-short form stable complexes at ERES, to jointly fulfill their roles in the secretion of bulky cargoes such as procollagens, pre-chylomicrons, and large pre-very low-density-lipoproteins (*Bosserhoff et al., 2003*; *Maeda et al., 2016*; *Malhotra and Erlmann, 2011*; *Malhotra and Erlmann, 2015*; *Saito et al., 2009*; *Saito et al., 2011*; *Santos et al., 2016*; *Wilson et al., 2011*).

At ERES TANGO1 assembles into rings that enclose COPII coats and create a sub-compartment dedicated to sorting, packing and exporting collagens (*Raote and Malhotra, 2019*; *Raote et al., 2018*; *Raote et al., 2017*). TANGO1's SH3-like domain binds collagens via the collagen-specific chaperone HSP47 (heat shock protein 47) in the ER lumen (*Ishikawa et al., 2016*). This binding of TANGO1 to HSP47-Collagen is proposed to trigger binding of its PRD to Sec23 in the cytoplasm. TANGO1's CC1 domain, that contains a subdomain named TEER (tether of ER Golgi intermediate compartment at ER), recruits ERGIC-53 membranes, which fuse with the nascent vesicle bud initiated by COPII inner coats (Sec23/Sec24) to grow the collagen filled container into an export conduit (*Raote and Malhotra, 2019*; *Raote et al., 2018*; *Santos et al., 2015*). Subsequent to collagen packing into this conduit, TANGO1 dissociates from HSP47 and collagen. TANGO1 is retained at ERES while collagens move forward in the anterograde direction (*Raote and Malhotra, 2019*).

The discovery of TANGO1 has made the process by which cells organize ERES and export collagen amenable to molecular analysis. We now describe the first human *TANGO1* mutation associated with a novel autosomal-recessive syndrome. These findings underscore the importance of TANGO1 in human (patho)physiology.

## Results

### Clinical description

Four brothers with a similar combination of congenital anomalies two of whom have already been described by *Cauwels et al. (2005)* were referred for oral examination to the Centre for Special Care, Ghent University Hospital, at the ages of 7 (*Figure 1B*; II.1;*1988), 3 (II.2;*1990), 6 (II.4;*2006), and 4 (II.5;*2008) years, and were followed up until present. Their parents are of Turkish origin and first cousins. The sister (II.3) as well as both parents (I.1 and I.2) were phenotypically normal. All four brothers presented with severe dentinogenesis imperfecta in both primary and permanent dentitions, delayed eruption of the permanent teeth, growth retardation, proportionate short stature, clinodactyly of the fifth finger, brachydactyly, platyspondyly, primary obesity, insulin-dependent diabetes mellitus (<1 IU of insulin/kg/day), sensorineural hearing loss, and mild intellectual disability (ID). Additional facultative symptoms included scoliosis, retrognathia, mild retinopathy, osteopenia, early onset periodontitis with premature tooth loss, hydronephrosis, and microalbuminuria (*Figure 1C*; *Table 1*).

### Whole exome sequencing (WES) revealed the disease-causing mutation in *TANGO1*

WES was performed in the four affected bothers and their parents. After filtering, 10 variants were found to be homozygous in all affected children and heterozygous in both parents (*Supplementary file 1*). In-depth data analysis revealed a synonymous variant in exon 8 of *TANGO1* (NM_001324062.1: c.3621A > G) as the most likely disease-causing mutation. *TANGO1* is known to be crucial for the secretion of collagens, consistent with phenotypes in our patients. WES results were validated by Sanger sequencing (*Figure 2A*). This *TANGO1* mutation was not present in large population databases such as ExAC or gnomAD (*Lek et al., 2016*). Although it does not alter the amino acid at the respective position (p.(Arg1207=)), the A > G substitution was predicted by ESE-finder to disrupt an exon splice enhancer (ESE) motif recognized by the human SR protein SC35 (*Figure 2—figure supplement 1A*). The mutation affects residue 1207 between the intramembrane and the CC1 domain within the cytoplasmic portion of TANGO1 (*Figure 1A*).

**Table 1.** Clinical symptoms in four affected brothers

| | II.1 | II.2 | II.4 | II.5 |
|---|---|---|---|---|
| Dentinogenesis imperfecta | x | x | x | x |
| Delayed eruption of permanent teeth | x | x | x | x |
| Juvenile periodontitis with early tooth loss | x | x | | |
| Growth retardation | x | x | x | x |
| Proportionate short stature | x | x | x | x |
| High nasal bridge | x | x | x | x |
| Retrognathia | | | | x |
| Phalangeal brachydactyly of fingers | x | x | x | x |
| Clinodactyly of 5th finger | x | x | x | x |
| Cone-shaped epiphyses in the hands | x | x | | x |
| Brachydactyly of toes | | | x | x |
| Platyspondyly (flattened vertebral corpora) | x | x | x | x |
| Scoliosis | | | x | x |
| Prominent knees | x | x | x | x |
| Mild intellectual disability | x | x | x | x |
| Sensorineural hearing loss | x | x | x | x |
| Mild retinopathy | x | | | x |
| Insulin-dependent diabetes mellitus | x | x | x | x |
| Primary obesity | x | x | x | x |
| Early onset puberty | | | | x |
| Pruritus | x | x | x | x |
| Asthma | x | x | x | x |
| Osteopenia | x | x | | |
| Hydronephrosis (junctional stenosis) | | | x | |
| Nephropathy (microalbuminuria) | | x | | |

Homozygosity mapping of two affected brothers (II.1 and II.2) identified a shared homozygous interval of ~19 Mb on chromosome 1, spanning GRCh37/hg19 coordinates 214,413,099–233,429,284 (rs12736101-rs6656327), including *TANGO1* and 28 disease-associated OMIM genes (*Figure 2—figure supplement 1B,C*). Apart from *TANGO1*, none of the shared homozygous intervals was endowed with a pathogenic mutation. In addition, no potentially disease-causing copy number variation (CNV) was detected by array comparative genomic hybridization (CGH).

## The *TANGO1* mutation leads to exon eight skipping by disrupting an exon splice enhancer both in vivo and in vitro

In order to investigate possible effects of the identified *TANGO1* mutation on pre-mRNA splicing, blood samples of the whole family were used for RNA isolation and reverse transcription PCR. Subsequent gel electrophoresis and cDNA sequencing revealed two splice products in the homozygous sons and their heterozygous parents, one representing the full length transcript being more abundant in the parents and another one lacking the entire exon eight being more abundant in the affected sons (*Figure 2B,C*; *Figure 3*). Exon eight skipping during *TANGO1* pre-mRNA splicing causes a frameshift and a premature stop codon 27 bp downstream of exon 7 (c.3610_3631delinsTCACGGAACAGCAAATTTCTGAGAAGTTGA) (*Figure 2D*). The predicted truncated protein lacks almost the entire cytoplasmic portion including CC1/TEER, CC2, and PRD (*Figure 1A*).

A minigene assay (*Figure 2—figure supplement 2*) was performed to confirm that the identified *TANGO1* mutation is sufficient to induce exon eight skipping. Transfection of cultured HEK293T cells

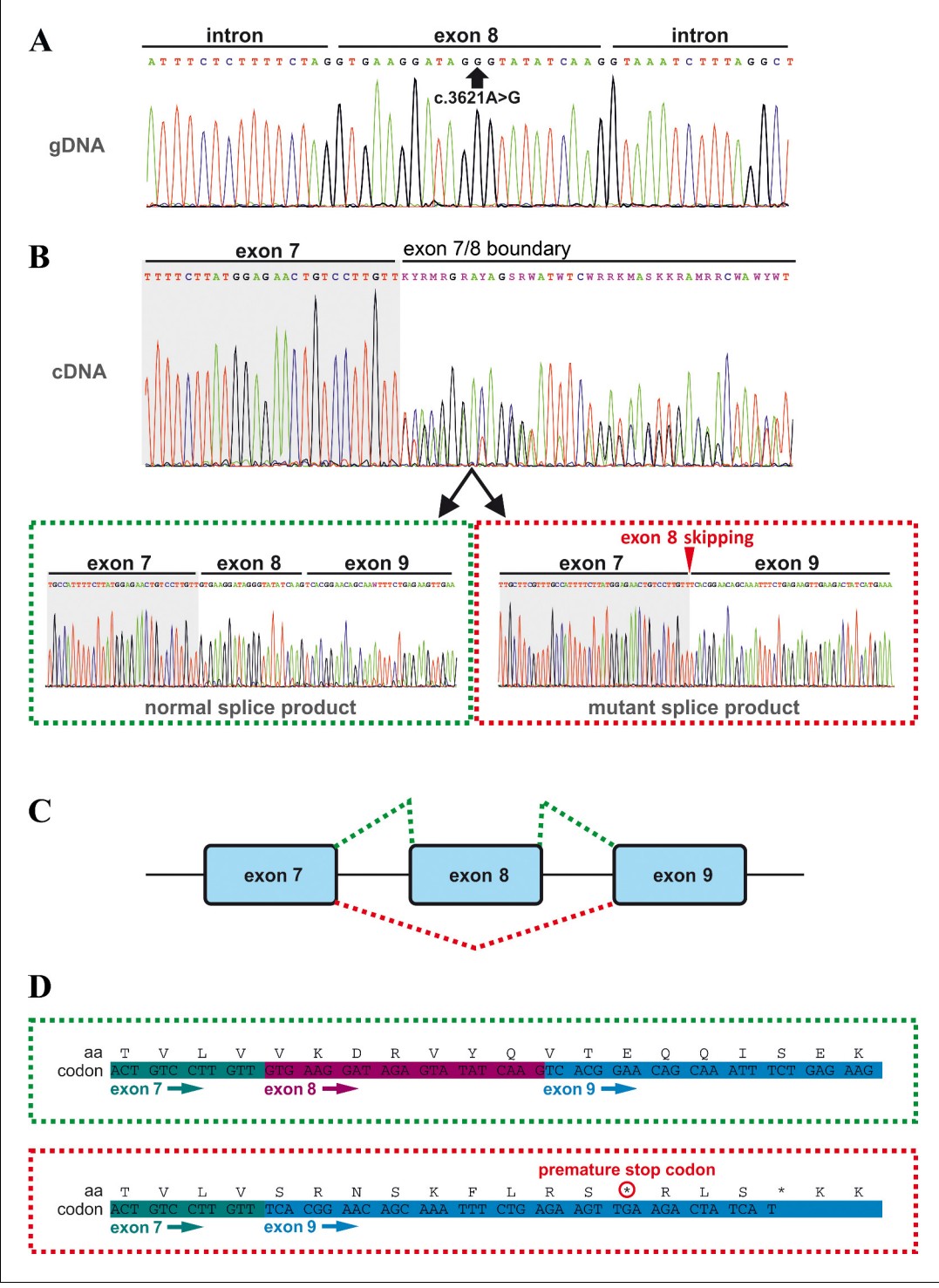

**Figure 2.** Effects of the *TANGO1* (c.3621A > G) mutation on pre-mRNA splicing. (**A**) The synonymous variant, which was identified by WES and validated by Sanger sequencing in all family members, resides in exon 8 of *TANGO1* at genomic position 222,822,182 (GRCh37/hg19). It is predicted to disrupt an exon splice enhancer (ESE) motif recognized by the human SR protein SC35. (**B**) Electropherograms of the *TANGO1* cDNA sequence of one affected child. Note the splitting of the sequence starting at the exon 7/8 boundary. Sequencing of individual bands after gel electrophoretic separation revealed *TANGO1* wild-type cDNA and cDNA lacking exon 8 (c.3610_3631delins30). For the cDNA sequencing of TANGO1 splice products, at least two (up to 6) technical replicates were performed for each family member. (**C**) Schematic representation of the alternatively used splice sites resulting in the normal *TANGO1* mRNA (green dotted lines) and in exon eight skipping (red dotted lines). (**D**)
*Figure 2 continued on next page*

*Figure 2 continued*

Consequences of *TANGO1* exon eight skipping on the reading frame and the amino acid level. Exclusion of exon eight causes a premature stop codon.

The online version of this article includes the following figure supplement(s) for figure 2:

**Figure supplement 1.** Sequence analysis.
**Figure supplement 2.** Minigene assay.
**Figure supplement 3.** Possible mechanisms underlying TANGO1 exon eight skipping.

with the Mnt vector carrying the mutated (c.3621A > G) *TANGO1* exon 8 and 250 bp flanking intronic sequences mainly produced vector-derived splice products lacking exon 8.

The observed splicing error could be due to disruption of an ESE motif recognized by the SR protein SC35 or to formation of a splice repressor motif recruiting the heterogeneous nuclear ribonucleoprotein A1 (hnRNP A1). To test this, cultured HeLa cells were transfected with either a wild-type or a mutated *TANGO1* vector (*Figure 2—figure supplement 3*) and then treated with a customized antisense oligonucleotide (*vivo* morpholino) targeting the entire *TANGO1* exon 8. SR proteins are required for proper exon inclusion during splicing and their absence can lead to exon skipping. The observed effects of the morpholino treatment on *TANGO1* splicing support the idea that the c.3621A > G mutation interferes with SR protein binding.

## The homozygous *TANGO1* mutation results in exon eight skipping in most splice products

Quantitative real-time (qRT) PCR on blood cDNA samples was performed to quantify the amount of mutant and normal *TANGO1* splice products, respectively, in homozygous and heterozygous mutation carriers compared to a normal control (*Figure 3*). The affected children consistently displayed the lowest amounts of normal splice product (mean RQ value: 0.39) and the highest amounts of the exon eight skipped product (mean RQ value: 4.58). Both parents showed more normal splice product (mean RQ value: 0.59) than their children but still only approximately half of that of the control. In contrast to the control individual, both parents also displayed a considerable proportion of the exon eight skipped splice product (mean RQ value: 2.66). Because of a processed transcript (ENST00000495210.1) without exon eight which is probably co-amplified by reverse transcription PCR, the exact ratios of the mutant versus the normal splice product could not be determined.

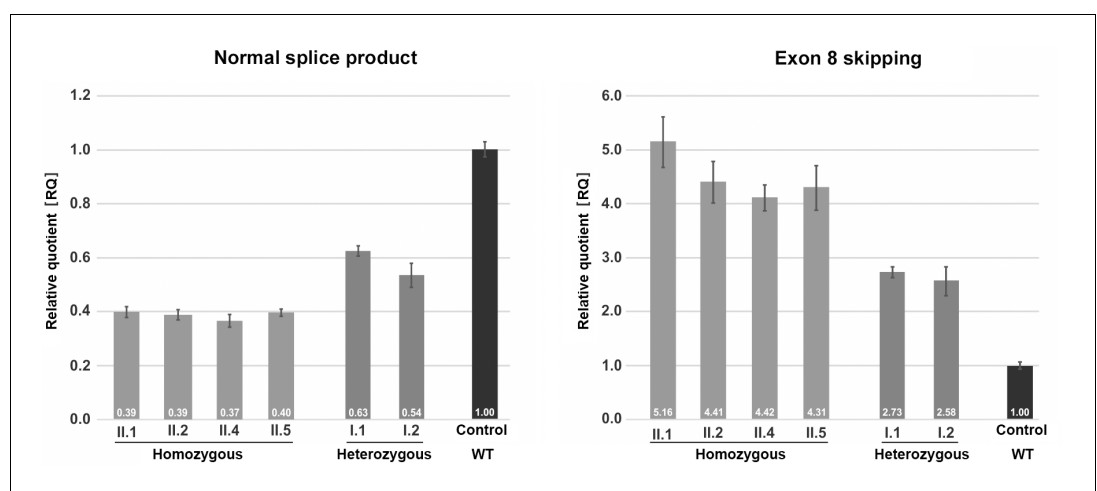

**Figure 3.** Quantification of *TANGO1* splice products in homozygous and heterozygous mutation carriers, compared to a control individual (without mutation). The right bar diagram shows the relative amounts of the normally spliced *TANGO1* cDNA and the left diagram of splice products lacking exon 8. The standard deviation of each bar represents the results of triplicate measurements. A control cDNA sample was used for normalisation and relative comparison (RQ = 1). By qRT-PCR the control sample used was representative for three other control individuals.

Unfortunately, it was not possible to design specific primers for the aberrant exon eight skipped splice product. In a homozygous state, the *TANGO1* c.3621A > G mutation leads to exon eight skipping in most splice products, whereas in the heterozygous parents the normal splice product is more abundant.

## The truncated TANGO1 protein does not localise to ER exit sites

To test the properties of the truncated TANGO1 protein, TANGO1 lacking exon 8 (Ex8-HA) was expressed in cultured U2OS cells. Cells were transiently transfected with cDNA for either wild-type

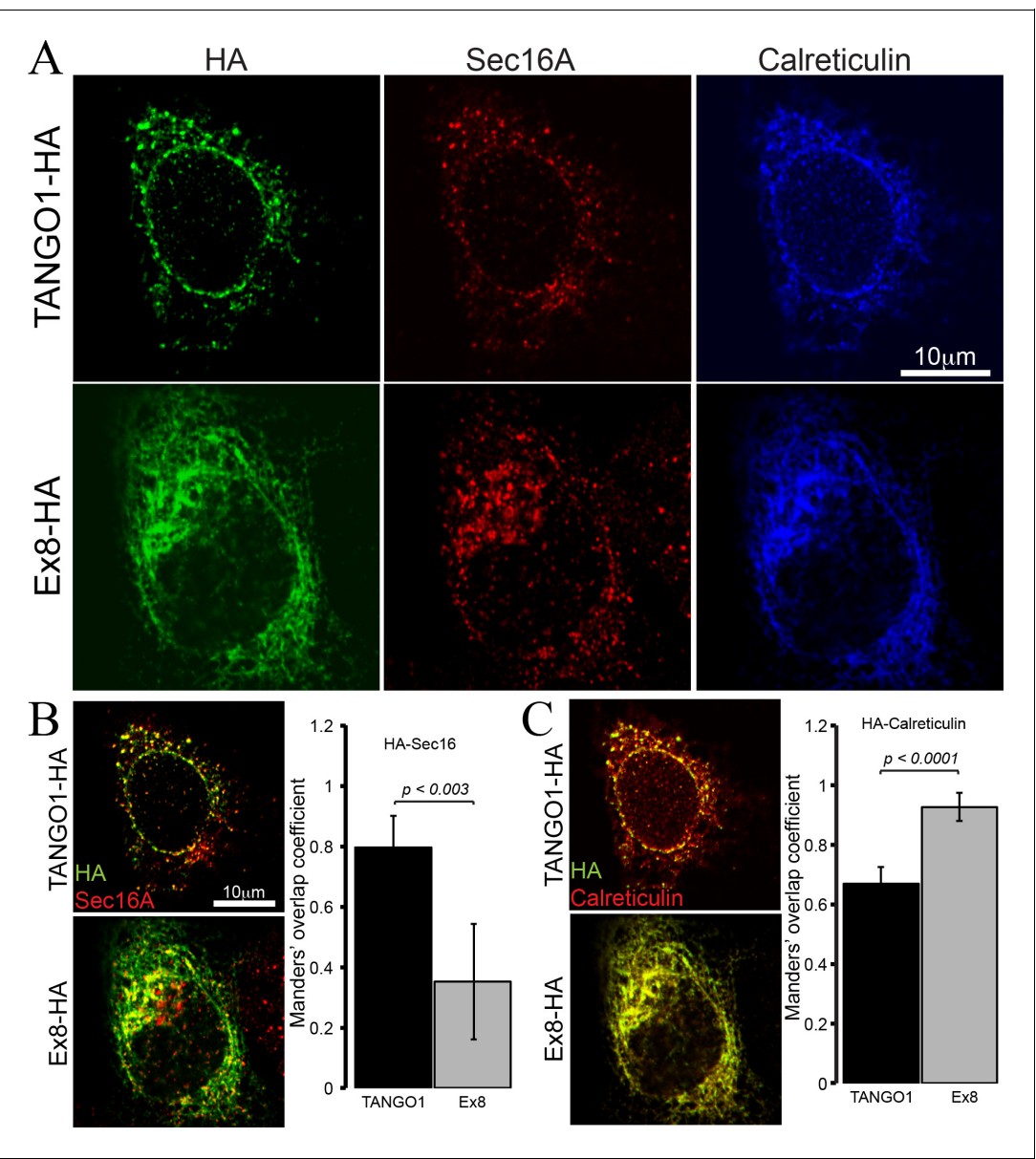

**Figure 4.** Ex8 mutant does not localise to ER exit sites. Immunofluorescence images of U2OS cells, transiently transfected with WT TANGO1-HA or Ex8-HA. Representative images of three independent experiments. (**A**) Cells were probed with anti-HA (green), anti-Sec16A (red) and anti-Calreticulin (blue) antibodies. Scale bar 10 μm. (**B**) Merged images of TANGO1-HA or Ex8-HA (green) and sec16A (red) and a plot comparing Manders' overlap coefficient of HA with sec16A in TANGO1- or Ex8-expressing U2OS cells. (**C**) Merged images of TANGO1-HA or Ex8-HA (green) with calreticulin (red) and a plot comparing Manders' overlap coefficient of HA and calreticulin in TANGO1- or Ex8-expressing U2OS cells. Student's t test was performed to compare the Manders' overlap coefficients, p values are shown.

TANGO1-HA or Ex8-HA (*Figure 4*). Forty-eight hours after transfection, cells were fixed, permeabilised and immunostained for HA (green), the ERES marker Sec16A (red) and the ER marker calreticulin (blue) (*Figure 4A*). WT TANGO1-HA (green) expressed in distinct puncta, which colocalised with Sec16A (red). On the other hand, Ex8-HA (green) was distributed in a more diffused pattern throughout the cell and it did not localise to ERES (red) (*Figure 4B*). WT TANGO1-HA (green) showed less association with calreticulin (red), while Ex8-HA (green) was almost entirely colocalised with calreticulin (red) (*Figure 4C*). These data are consistent with our understanding of TANGO1 function, as its cytoplasmic domains are required to recruit TANGO1 to ERES. The Ex8 mutant lacks any cytoplasmic domains and consequently is distributed through the ER.

## Cells expressing the truncated TANGO1 show reduced levels of intracellular and secreted collagen I

All affected individuals showed the highest amounts of the exon eight skipped splice product compared to the normal *TANGO1* splice product (*Figure 3*). However, it was not possible to determine how the relative abundance of the two splice products translated to protein levels in patient-derived samples. Therefore, possible effects of the overexpression of Ex8-HA on top of endogenous levels of normal *TANGO1* splice product at the cellular level were investigated using human osteosarcoma U2OS as a model system. U2OS cells produce and rapidly secrete collagen I, so they are the ideal system to monitor possible effects of Ex8-HA overproduction on collagen homeostasis. For this purpose, a stable U2OS cell line expressing Ex8-HA under a constitutive promoter was generated and compared with control cells by microscopy. Cells expressing Ex8-HA showed weaker and more diffuse staining of collagen I compared with control cells (*Figure 5—figure supplement 1A*). To confirm that this reduction was not due to unequal immunofluorescent staining, total RNA was isolated from the two cell populations and the relative amount of collagen I/GAPDH transcript was quantified by qRT-PCR analyses (*Figure 5—figure supplement 1B*). This confirmed that collagen I expression is reduced in the Ex8-HA stable cell line. The reduction in expression was not due to unfolded protein response (UPR) as there was no difference in XBP-1 splicing upon expression of Ex8-HA (*Figure 5—figure supplement 1C*).

The next step was to test whether the secretion of collagen I was also affected. Since different rates of collagen synthesis will affect the amount available for secretion in the two cell populations, a cycloheximide chase experiment was performed (*Figure 5*). By inhibiting protein synthesis, it was possible to monitor the rate of secretion of the available pool of collagen I present in the cells at time zero. By quantifying the relative amount of collagen I present in the cells and in the media at each time point, a drastic reduction in the rate of collagen I secretion from the EX8-HA stable cell line compared to control cells was observed. Importantly, this effect was not due to a general reduction of protein secretion since the small cargo antitrypsin was produced and secreted at a comparable rate in the two cell populations. Under the same conditions, we tested secretion of collagen IV and collagen XII in EX8-HA stable line. Six independent trials revealed a consistent trend of a net reduction in their secretion, but because of the low relative abundance of these collagens the magnitude of the effect varied considerably and cannot be statistically quantified (*Figure 5—figure supplement 2*).These secretion defects were specific to Ex8 expression as we observed no change in collagen I secretion when full-length TANGO1 was overexpressed in U2OS (*Figure 5—figure supplement 3*). Collectively, these results show that expression of the exon eight skipped splice product even in the presence of full length TANGO1 affects collagen I homeostasis.

## Discussion

Our study provides evidence that aberrant expression of a truncated TANGO1 protein and/or reduced levels of fully functional TANGO1 protein, or likely a combination of both causes a novel syndrome due to disturbances in cellular protein secretion. The heterozygous parents show exon eight skipping but no detectable symptoms, whereas all four homozygous children are severely and similarly affected. This is consistent with a threshold model, where the disease only manifests when the ratio of truncated versus normal protein exceeds a critical level. *Tango1* knockout mice represent a full loss-of-function (LoF) situation and are defective for the secretion of numerous collagens, exhibiting short-limbed dwarfism, compromised chondrocyte maturation and bone mineralization, and other features (*Wilson et al., 2011*), resembling our patients' phenotype. The human *TANGO1*

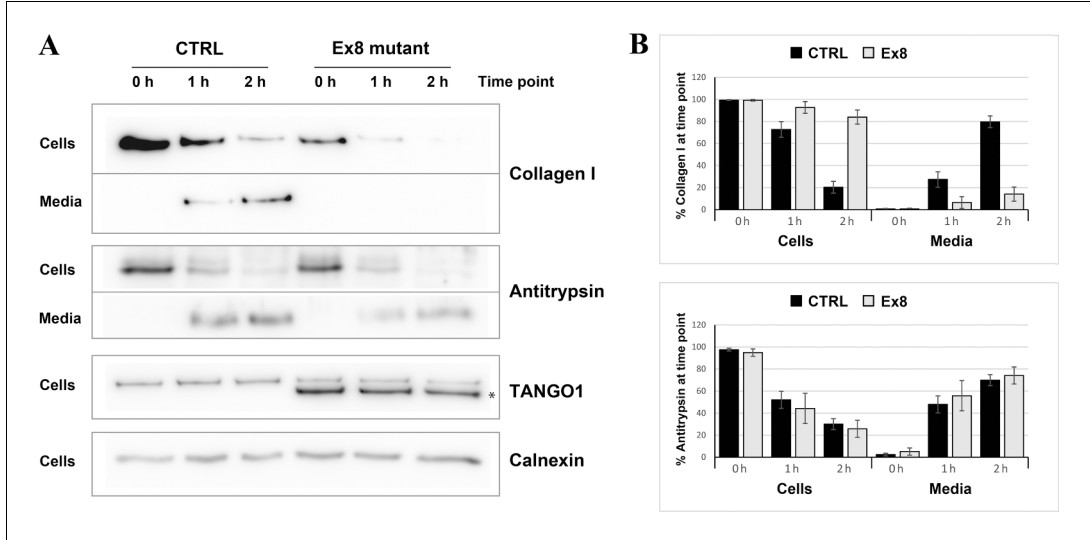

**Figure 5.** Ex8 mutant expression reduces collagen I secretion in U2OS cells. (A) Media of U2OS control cells (CTRL) or stably expressing Ex8-HA mutant (Ex8) were replaced with OptiMEM media containing 0.25 mM ascorbic acid and 50 μM cycloheximide to block protein synthesis and follow collagen secretion. Cell extracts and media were collected at the indicated time points and analysed by SDS-PAGE followed by Western blotting with antibodies raised against Collagen I, TANGO1, and Calnexin (loading control). (*) indicates Ex8-HA. Representative images of four independent experiments. (B) For each time point, the band intensities of collagen I (upper panel) or antitrypsin (lower panel) were measured for the cell extract and media samples and expressed as percentage of the total (cells plus media). Each graph represents the average quantification of four experiments and corresponding standard deviations.

The online version of this article includes the following figure supplement(s) for figure 5:

**Figure supplement 1.** TANGO1 exon eight mutant (Ex8) expression reduces collagen I expression in U2OS cells without inducing UPR.

**Figure supplement 2.** Ex8 mutant expression reduces collagen XII and collagen IV secretion in U2OS cells.

**Figure supplement 3.** TANGO1 full-length overexpression has no effect on collagen I secretion in U2OS cells.

locus lies within a homozygous interval shared among the affected children, which strengthens its role as the disease-causing gene in the investigated family. *TANGO1* does not seem to be haploin-sufficient, since there are several heterozygous LoF mutation carriers listed in big population data-bases, ExAC and gnomAD (*Lek et al., 2016*). However, no homozygous LoF mutation carriers have been reported so far. Thus, complete ablation of functional TANGO1 may cause embryonic lethality in humans.

The skipped exon eight in full length TANGO1 corresponds to exon three in the isoform TANGO1-short. Its exclusion there also leads to a premature stop codon 27 bp downstream of exon 2. TANGO1-short has a similar structure to TANGO1, but lacks the lumenal portion within the cargo-binding SH3 domain. However, TANGO1-short has been shown to substitute the function of TANGO1 in collagen export, and vice versa (*Maeda et al., 2016*). It has been postulated that the cytoplasmic portion's capacity to recruit ERGIC-53 membranes, Sec23/24 complexes, and cTAGE5, shared by both isoforms, is sufficient to export collagen at the ER (*Saito et al., 2009*; *Santos et al., 2015*). The lumenal SH3 domain may therefore rather play a role in modulating the efficiency or the quality (folding status) of collagens to be secreted (*Maeda et al., 2016*; *Raote et al., 2018*; *Saito et al., 2009*). Since the identified mutation in exon eight does not only compromise the func-tion of TANGO1, but most likely also of TANGO1-short, the short isoform cannot attenuate our patients' phenotype.

The truncated TANGO1 protein is unable to access ERES and is distributed throughout the ER (*Figure 4*). Its expression in cells along with endogenous full length TANGO1 displayed a negative effect on collagen secretion. This effect is specific to collagen, as the secretion of antitrypsin remains unaffected (*Figure 5*). In two previous descriptions of TANGO1 function (*Nogueira et al., 2014*; *Saito et al., 2009*), we reported that it had no obvious role in collagen I export. However, it has

been demonstrated subsequently that TANGO1 functions to export all soluble collagens tested thus far, including collagen I. We have noticed a change in the mRNA level of collagen α1 chain in U2OS cells expressing mutant TANGO1. Although, we have not tested the effects on the expression of other chains of collagen one and other collagens, it is conceivable that expression of this mutant causes accumulation of collagens in the ER, which then affect their further synthesis by a feedback mechanism. However, this appears not to involve the activation of UPR in U2OS cells. Whether this is also the situation in vivo in the individuals with TANGO1 mutation is unknown.

Type I collagens provide tensile strength to connective tissue and are abundant in bone, skin, dentin, cementum, tendons, and ligaments (*Deshmukh et al., 2016*). Impaired collagen I secretion may underlie several key symptoms in our patients, in particular the tooth and skeletal abnormalities. Haploinsufficiency or gain of function mutations in *COL1A1* and *COL1A2*, coding for the pro-collagen I chains, can cause different forms of osteogenesis imperfecta (OI) with (OMIM #259420) or without dentinogenesis imperfecta (DGI). Hearing loss is commonly found in OI patients. Mutations in other collagen genes, for example *COL2A1*, *COL4A3*, *COL4A4*, *COL4A5*, *COL4A6*, *COL9A1*, *COL9A2* and *COL9A3*, *COL11A1*, and/or *COL11A2* have been associated with hereditary hearing loss (https://hereditaryhearingloss.org/). TANGO1 may control secretion of cargoes other than collagens. This is particularly noteworthy based on the recently reported role of TANGO1 in mucin export in *Drosophila* salivary glands and insulin secretion (*Fan et al., 2017*; *Kang et al., 2019*; *Reynolds et al., 2019*) Therefore, the complex phenotype of our patients, including DGI, diabetes, intellectual disability and hearing impairment, but no brittle bones may at least partially be due to aberrant export and secretion of other proteins in a tissue specific manner.

For example, mutations in the dentin sialophosphoprotein (*DSPP*) gene and, by extrapolation, TANGO1-associated defects in DSPP secretion, may cause DGI 1 (#605594) with or without hearing loss (*Xiao et al., 2001*) as well as DGI, Shields type II (#125490) and III (#125500), which may be phenotypic variation of the same entity rather than separate diseases (*Kim et al., 2005*).

Defective secretion of other molecules may cause diabetes mellitus and pubertas praecox. Endocrinological examination revealed that the glucose intolerance of the affected children is due to reduced levels of secreted insulin. cTAGE5 is known to cooperate with TANGO1 in the mega cargo secretion pathway (*Saito et al., 2011*), but has also been shown to play a pivotal role in ER to Golgi trafficking of small molecules like proinsulin (*Fan et al., 2017*). The knockout of *cTAGE5* in pancreatic β-cells resulted in defective islet structure, reduced insulin secretion, and severe glucose intolerance in mice (*Fan et al., 2017*). Additionally, a correlation between TANGO1 phosphorylation and proinsulin trafficking in mouse pancreatic β-cells has recently been discovered (*Kang et al., 2019*). In this light, it will be intriguing to investigate the role of TANGO1 in the insulin secretion pathway in future studies.

Collectively, the investigated family presents the first *TANGO1*-associated syndrome in humans, highlighting the role of fully functional TANGO1 in various disease pathways and bringing new potential target molecules of TANGO1 into focus.

## Materials and methods

### Whole exome sequencing

Exome capture was performed according to the Illumina Nextera Rapid Capture Enrichment library preparation (individuals II.1 and II.2) or the Illumina TruSeq Rapid Exome library preparation kit (individuals I.1, I.2, II.4, and II.5), using 50 ng of genomic DNA. Paired-end sequencing of the libraries was performed with a NextSeq500 sequencer and the v2 reagent kit (Illumina, San Diego, California, USA). Sequences were mapped to the human genome reference (NCBI build37/hg19 version) using the Burrows-Wheeler Aligner. Aligned reads ranged between 82,649,383 and 102,537,469. The mean coverage was ≥52 with 90.3% of the exome being covered at least 10x. A total of 237,330–297,312 variants per sample were called and analyzed using GensearchNGS software (PhenoSystems SA, Braine le Chateau, Belgium). Variants with a coverage of ≤20, a Phred-scaled quality of ≤15, a frequency of ≤20, and a MAF of ≥1% were neglected. Two control samples from healthy individuals were used for filtering out platform artefacts. Alamut Visual (Interactive Biosoftware, Rouen, France) software including prediction tools like SIFT, MutationTaster, and PolyPhen-2 was used for variant prioritization. Potential effects of a variant on pre-mRNA splicing were evaluated by

SpliceSiteFinder-like, MaxEntScan, NNSPLICE, GeneSplicer, Human Splicing Finder, ESEfinder, RESCUE-ESE, and EX-SKIP. Population databases like ExAC, gnomAD, and GME revealed whether a variant has been previously found. Protein expression, structure, and functional aspects were investigated with UniProt and The Human Protein Atlas. Information on mouse models was retrieved from the MGI database.

## Sanger sequencing

*TANGO1* exon eight was amplified by a touchdown PCR program using primers in the flanking introns (forward 5'-TCAGACCACAACATATCACTACTGG-3'; reverse 5'-TACTCTATCATACAACCTGGCAACC-3'). A clean-up step with ExoSAP-IT (Applied Biosystems, Foster City, California, USA) was followed by the sequencing reaction using the BigDye Terminator Cycle Sequencing Kit v1.1 (Applied Biosystems). Sequencing was conducted on a 3130XL capillary sequencer (Applied Biosystems) and data analysis was performed with Gensearch (PhenoSystems SA).

## Microarray analyses

Two affected children (II.1 and II.2) were genotyped by Life and Brain (Bonn, Germany), using the Infinium Global Screening Array-24 v1.0 BeadChip (Illumina). Shared homozygous intervals were identified with HomozygosityMapper (*Seelow et al., 2009*).

Array CGH was performed using the CGX DNA labeling kit (PerkinElmer, Waltham, Massachusetts, USA) and the CGX-HD array (PerkinElmer) that covers clinically relevant regions with 180,000 oligonucleotide marker. A male genomic DNA sample served as a reference. The hybridized array was scanned with the NimbleGen MS 200 Microarray Scanner (Roche, Basel, Switzerland). Data analysis was conducted with CytoGenomics 2.5 (Agilent Technologies, Santa Clara, California, USA) and Genoglyphix 3.0 (PerkinElmer) software using annotations from GRCh37/hg19.

## Minigene assay

To investigate possible effects of the *TANGO1* mutation on pre-mRNA splicing, the homozygous individual II.1 was compared to a normal control sample in a minigene assay. The plasmid pSPL3b-cam vector (*Burn et al., 1995*) is endowed with a chloramphenicol resistance, an SV40 promoter, SD6 and SA2 primer sequences, as well as a multiple cloning site including recognition sites for *XhoI* and *BamHI* (*Figure 2—figure supplement 2A*). A 521 bp amplicon including *TANGO1* exon eight and ~250 bp flanking intronic sequences was generated from genomic DNAs of the patient and a control, using primers with recognition sites for *XhoI* and *BamHI* at their 5' ends (fwd 5'-AATTC TCGAGTATCTTTAGCTGTGCAAAGT-3'; rev 5'-ATTGGATCCAAGGTCAATCTGCCCCAAAT-3') and the Q5 High-Fidelity DNA Polymerase (New England Biolabs, Ipswich, Massachusetts, USA). The PCR products were purified with the GenElute PCR Clean-Up kit (Sigma-Aldrich, St. Louis, Missouri, USA), digested with *XhoI* and *BamHI* in CutSmart Buffer (New England Biolabs), again purified, and finally ligated into the linearized vector, using T4 DNA Ligase and T4 DNA Ligase Reaction Buffer (New England Biolabs).

Vector constructs were transformed into DH5α bacteria by heat shock for 90 s at 42°C and then plated onto LB/agar/chloramphenicol Petri dishes. Following overnight incubation at 37°, a colony screen was performed using SD6 (fwd 5'-TCTGAGTCACCTGGACAACC-3') and the *TANGO1* exon eight reverse primer (see above). Positive clones (with insert) were cultured overnight and the vector constructs extracted using the GenElute Plasmid Miniprep kit (Sigma-Aldrich). Sequencing was performed with 100 fmol of vector constructs, SD6 forward and *TANGO1* exon eight reverse primers. Three vector constructs were selected for splicing experiments, one with the wild-type *TANGO1* exon eight and flanking sequences (WT vector) and another one with the mutated *TANGO1* exon 8 (Mnt vector). The pSPL3b-cam vector without insert (CTRL) served as control.

Aliquots of $4 \times 10^{15}$ HEK293T cells were plated into 6-well-plates and transfected with vector constructs (WT, Mnt, or CTRL) using the FuGENE HD Transfection Reagent (Roche). After 24 hr of incubation at 37°C, RNA was isolated with the miRNeasy Mini kit (Qiagen, Venlo, Netherlands). cDNA was synthesized using the High Capacity RNA-to-cDNA kit (Applied Biosystems), amplified by a touchdown PCR program using SD6 forward and SA2 reverse (5'-ATCTCAGTGGTATTTGTGAGC-3') primers, purified with ExoSAP-IT (Applied Biosystems) and finally sequenced. The resulting splice products were compared by gel electrophoresis and cDNA sequencing. Since the Mnt vector

produced two separate *TANGO1* splice products, individual cDNA molecules were cloned using the TA Cloning Kit and the Dual Promoter (pCRII) protocol (Invitrogen, Carlsbad, California, USA).

## RNA isolation from whole blood samples, cDNA synthesis and sequencing

Peripheral blood samples were collected in PAXgene Blood RNA tubes and RNA was isolated using the PAXgene Blood RNA Kit (PreAnalytiX, Hombrechtikon, Switzerland). Reverse transcription was performed with the High Capacity RNA-to-cDNA Kit (Applied Biosystems). cDNA was amplified by a touchdown PCR program using primers located in *TANGO1* exon 6 and 11, respectively (fwd 5'-ACACTCCTATGGATGCTATTGATGC-3'; rev 5'-CTCTCTCAGATTCTAGCATAACACG-3'). After a clean-up step with ExoSAP-IT (Applied Biosystems), cDNA sequencing was performed with the Big-Dye Terminator Cycle Sequencing Kit v1.1 (Applied Biosystems) and a 3130XL capillary sequencer (Applied Biosystems). Data analysis was performed with Gensearch (PhenoSystems SA) and Codon-Code Aligner (CodonCode Corporation, Centerville, Massachusetts, USA). Multiple PCR products were separated by gel electrophoresis. cDNA of individual cut out bands was isolated with the QIA-quick Gel Extraction Kit (Qiagen) and 1–3 µl of gel extracts were used for sequencing. At least two technical replicates were performed for each analyzed individual.

## Morpholino assays

Effects of different *vivo* morpholinos on *TANGO1* pre-mRNA splicing were tested on HeLa cells with a different genetic background. $2 \times 10^5$ HeLa cells in 2 ml DMEM (Sigma-Aldrich) each were plated into the required number of wells of a 6-well-plate and incubated at 37°C for 24 hr. Depending on the research question, cells were then transfected with the *TANGO1* WT or Mnt vector, or used without vector transfection for morpholino treatment. For transfection, 2 µg vector were suspended in up to 95 µl DMEM and mixed with 6 µl FuGENE HD Transfection Reagent (Roche). The mixture was incubated at room temperature for 15 min and then slowly added to the cells.

The customized TGO morpholino (TACCTTGATATACTCTATCCTTCAC) targets the entire *TANGO1* exon 8. A standard control morpholino (GeneTools, Philomath, Oregon, USA) was used to exclude unspecific effects. The *vivo* morpholinos were added at a final concentration of 5 or 7 µM. After 24 hr incubation, the medium was removed, cells were washed with 2 ml 1x PBS, detached with 0.5 ml Trypsin-EDTA Solution (Sigma-Aldrich) and then transferred into 2 ml tubes with 1x PBS. Samples were centrifuged at 3000 g for 5 min at 4°C. Supernatant was removed and RNA isolated using the miRNeasy Mini Kit (Qiagen). cDNA synthesis and sequencing were performed as described above, using vector-specific primers (SD6 and SA2). This ensured that only vector-derived splice products were analysed.

## Quantitative real-time PCR and XBP-1 splicing

qRT-PCRs were performed to obtain a relative ratio of either the exon eight skipped *TANGO1* or the normal splice product for homozygous and heterozygous mutation carriers, compared to a reference sample, which was found to be representative for normal individuals (without mutation). Two housekeeping genes, *HPRT1* (fwd 5'-TGACACTGGCAAAACAATGCA-3'; rev 5'-GGTCCTTTTCAC-CAGCAAGCT-3') and *IPO8* (fwd 5'-CGAGCTAGATCTTGCTGGGT-3'; rev 5'-CGCTAA TTCAACGGCATTTCTT-3') served as endogenous controls. Assay 1 (fwd 5'-GACTGCCATGGAAACC TGTATT-3'; rev 5'-TCCGTGAAACAAGGACAGTTCT-3') exclusively amplified the mutant splice product where exon seven is followed by exon 9; assay 2 (fwd 5'-GACTGCCATGGAAACCTGTATT-3'; rev 5'-CCTTCACAACAAGGACAGTTCT-3') the normal splice products where exon seven is followed by exon 8. The PCR reaction consisted of 4 µl (10 ng) cDNA, 1 µl (2.5 pmol) primer pair, 2 µl 5x HOT FIREPol EvaGreen qPCR Mix Plus, and 3 µl water. All samples were run in technical triplicates. Cycling conditions on a ViiA 7 Real-Time PCR System (Applied Biosystems) were as follows: 95°C for 15 min, 40 cycles of 95°C for 15 s, 60°C for 20 s, and 72°C for 20 s. The melt curve was obtained from 60°C to 95°C and indicated no secondary amplicons.

In addition, qRT-PCRs were conducted to measure the relative expression of full length TANGO1, truncated TANGO1, and Collagen I. For this purpose, U2OS cells WT or stabling expressing Ex8-HA were lysed and total RNA extracted with the RNeasy extraction kit (Qiagen). cDNA from two biological replicates was synthesized with Superscript III (Invitrogen). Primers for COL1A1 (fwd 5´-GTGG

TCAGGCTGGTGTGATG-3′; rev 5′-CAGGGAGACCCTGGAATCCG-3′), TANGO1 lumenal portion (fwd 5′-TGGAAGTGTTGGACGCACTTTT-3′; rev 5′-TCAGGTTCAGGTTCCCTTTCCT-3′) or cytosolic portion (fwd 5′-CTCAGCTCTGCGGACCTTTT3′; rev 5′-GTGAACAGTCCTGGCTAGTGC-3′) were designed using Primer-BLAST (NCBI) (*Ye et al., 2012*) with the annealing temperature to 60℃. To determine expression levels of collagen I and the two forms of TANGO1, qRT-PCR was performed with Light Cycler 480 SYBR Green I Master (Roche) according to manufacturer's instructions. For each biological replicate, three technical replicates were used to determine mean values and standard deviations.

PCR amplification of *XBP-1* from cDNA was done as described previously using 5'-AAACAGAG TAGCAGCTCAGACTGC-3' and 5'-TCCTTCTGGGTAGACCTCTGGGAG-3' (*Calfon et al., 2002*). IRE1 couples endoplasmic reticulum load to secretory capacity by processing the *XBP-1* mRNA. PCR products were separated in 3% agarose gels. Fragment corresponding to unspliced version of *XBP-1* (473 bp) was cut upon PstI digestion (290 bp and 183 bp). Fragment corresponding to spliced product was not affected, since the PstI restriction site is lost after IRE1-mediated cleavage and splicing of the mRNA.

## Cell culture and transfection

HeLa, HEK293T and U2OS cells (obtained from the eukaryotic cell line collection maintained by the Centre for Genomic Regulation, Spain) were grown at 37℃ with 5% $CO_2$ in complete DMEM with 10% FBS. For lentiviral infection of Ex8-HA into U2OS cells, lentiviral particles were produced by cotransfecting HEK293 cells with pHRSIN/Ex8-HA or pHRSIN/TANGO1-HA plasmid and a third-generation packaging vector pool using TransIT-293 (Mirus Bio, Madison, Wisconsin, USA). 72 hr after transfection, the viral supernatant was harvested, filtered, and directly added to U2OS cells. Infected cells were selected using 500 µg/ml hygromycinB (Invitrogen). All cell lines were confirmed to be free of mycoplasma contamination on a monthly basis.

## Collagen-secretion assays

The media of U2OS cells was replaced with OptiMEM medium (Thermo Fisher Scientific, Waltham, Massachusetts, USA) containing 0.25 mM ascorbic acid and 50 µM cycloheximide (Sigma-Aldrich) for up to 2 hr to allow for collagen secretion. The media were collected at 0, 1, and 2 hr time points, centrifuged at low speed to remove any cells or cellular debris, and the supernatants were denatured at 65℃ for 10 min with Laemmli SDS sample buffer. For cell extracts, cells were washed with PBS, lysed in buffer A (50 mM Tris-Cl, pH7.4, 150 mM NaCl, 1 mM EDTA, 1% Triton X-100) plus proteases inhibitors (Roche), and centrifuged at 14,000 rpm for 15 min at 4℃. The supernatants were denatured at 65℃ for 10 min with Laemmli SDS sample buffer. Media and cell lysate were subjected to SDS-PAGE (6% or 8% acrylamide) and Western blotting with antibodies raised against collagen I, collagen XII, collagen IV, TANGO1, calnexin, hemagglutinin (HA), and antitrypsin. Band intensities were measured using QuantityOne (Bio-Rad, Hercules, California, USA), and four independent repetitions of the experiment were used to plot the graph. For each time point, the band intensities of collagen I or antitrypsin were measured for the cell extract and media samples, and expressed as percentage of the total (cells plus media). Each graph represents the average quantification of four experiments and corresponding standard deviations.

## Immunofluorescence staining

Cells grown on coverslips were fixed with cold methanol for 10 min at −20℃ and incubated with blocking reagent (Roche) for 30 min at RT. Primary antibodies were diluted in blocking reagent and incubated overnight at 4℃. Secondary antibodies conjugated with Alexa Fluor 488, 594 or 647 (Invitrogen) were diluted in blocking reagent and incubated for 1 hr at room temperature. Images were taken with a TCS SP8 or TCS SPE confocal microscope (Leica Microsystems, Wetzlar, Germany) with a 63 × objective. Images processing was performed with ImageJ. Images are representative of three independent experiments. Manders' overlap coefficient was calculated using ImageJ plugin JACoP. Student's t test was performed to compare the Manders' overlap coefficients.

## Antibodies

Antibodies used in Western blotting and immunofluorescence microscopy were as follows: TANGO1 (Sigma-Aldrich); beta-tubulin (Sigma-Aldrich); calreticulin (Novus Biologicals, Centennial, Colorado, USA); Calnexin (Abcam, Cambridge, United Kingdom); antitrypsin Ab-1 (NeoMarkers, Fremont, California, USA); collagen I and collagen IV (Abcam, Cambridge, United Kingdom); collagen XII (Santa Cruz Biotechnology, Dallas, Texas, USA); Sec16A (Sigma-Aldrich); rat hemagglutinin (Roche) or mouse hemagglutinin (Santa Cruz Biotechnology, Dallas, Texas, USA).

## Acknowledgements

We would like to thank the family for their participation.

## Additional information

### Competing interests

Vivek Malhotra: VM is Senior Editor of Elife. The other authors declare that no competing interests exist.

### Funding

| Funder | Grant reference number | Author |
| --- | --- | --- |
| Ministerio de Economía y Competitividad | SEV-2012-0208 | Ombretta Foresti Ishier Raote Vivek Malhotra |
| Ministerio de Economía y Competitividad | BFU2013-44188-P | Ombretta Foresti Ishier Raote Vivek Malhotra |
| Ministerio de Economía y Competitividad | CSD2009-00016 | Ombretta Foresti Ishier Raote Vivek Malhotra |
| Ministerio de Economía y Competitividad | IJCI-2017-34751 | Ishier Raote |
| Ministerio de Economía y Competitividad | RYC-2016-20919 | Ombretta Foresti |

The funders had no role in study design, data collection and interpretation, or the decision to submit the work for publication.

### Author contributions

Caroline Lekszas, Validation, Investigation, Visualization, Methodology, Writing - original draft; Ombretta Foresti, Validation, Investigation, Visualization, Methodology, Writing - review and editing; Ishier Raote, Supervision, Validation, Investigation, Visualization, Methodology, Writing - review and editing; Daniel Liedtke, Eva-Maria König, Barbara Vona, Supervision, Methodology; Indrajit Nanda, Supervision, Investigation; Peter De Coster, Rita Cauwels, Resources, Investigation; Vivek Malhotra, Thomas Haaf, Conceptualization, Supervision, Project administration, Writing - review and editing

### Author ORCIDs

Caroline Lekszas  http://orcid.org/0000-0003-4074-3776
Ombretta Foresti  https://orcid.org/0000-0002-6878-0395
Ishier Raote  https://orcid.org/0000-0002-5898-4896
Daniel Liedtke  https://orcid.org/0000-0003-0934-7169
Barbara Vona  http://orcid.org/0000-0002-6719-3447
Rita Cauwels  http://orcid.org/0000-0001-7615-5621
Vivek Malhotra  https://orcid.org/0000-0001-6198-7943
Thomas Haaf  https://orcid.org/0000-0002-0737-0763

## Ethics

Human subjects: Informed consent from affected individuals and/or their parents was obtained prior to initiating our investigation. Consent for publication of clinical data and genetic testing results was obtained from the affected individuals and/or their parents. This study was approved (205/11 and 46/15) by the Ethics Committee of University of Würzburg and was performed in accordance with the Declaration of Helsinki.

## Decision letter and Author response

Decision letter https://doi.org/10.7554/eLife.51319.sa1
Author response https://doi.org/10.7554/eLife.51319.sa2

# Additional files

## Supplementary files

• Supplementary file 1. Whole Exon Sequencing (WES) was performed in the four affected bothers and their parents. Shown are the 10 variants found to be homozygous in all affected children and heterozygous in both parents.

• Transparent reporting form

## Data availability

All data generated or analysed during this study are included in the manuscript and supporting files.

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
