## [Decision Letter]

**Acceptance summary:**

The human protein TANGO1 is ubiquitously expressed, localises to the endoplasmic reticulum (ER) exit site, binds various bulky cargo molecules including collagens and facilitates the export of cargoes from the ER by creating giant transport carriers. Malhotra and colleagues discovered and characterised TANGO1 in the past. In the present paper, Haaf, Malhotra and their colleagues identified a first patient with a balletic mutation in the TANGO1-encoding MIA3 gene, which leads to a truncation of the protein. Since the authors do not have access to patient cells, they established a surrogate cell model and characterised the relationship between the mutation and protein truncation, and the exocytosis defect of collagens. In light of the complex disease phenotype, the authors also discuss previously published data, which showed that the ubiquitously expressed TANGO1 is packing various cargo for exocytosis and support the complex disease syndrome of the patient. This paper highlights the importance of TANGO1 for human disease.

**Decision letter after peer review:**

Thank you for submitting your article "Biallelic TANGO1 mutations cause a novel syndromal disease due to hampered cellular collagen secretion" for consideration by *eLife*. Your article has been reviewed by three peer reviewers, including Reinhard Faessler as the Reviewing Editor and Reviewer #1, and the evaluation has been overseen by Suzanne Pfeffer as the Senior Editor. The following individual involved in review of your submission has agreed to reveal their identity: Karl E Kadler (Reviewer #3).

The authors identified patients with a homozygous TANGO1 gene mutation resulting in a truncated TANGO1 protein and a complex disease phenotype including an expected collagenopathy. The reviewers spent quite a bit of time reviewing and then discussing this paper, in part because some critical experiments are missing and corrections/explanations need to be included into a revised manuscript. Based on this discussion the Reviewing Editor has drafted this decision to help prepare a revised submission.

Essential revisions:

1) The TANGO1 mutation profoundly affects collagen I secretion, although a regulation of collagen I secretion was not observed in Saito et al. What is the explanation of this discrepancy?

2) Determine whether also the secretion of additional collagens, collagen II and VII are affected by the mutation, ideally in patient-derived fibroblasts.

3) Why were HeLa cells used in Figure 3 when U2OS cells would have been a better cell choice? HeLa are not a good cell choice for studies of collagen. Do HeLa cells express prolyl 4-hydroxylase, which is essential for procollagen folding and assembly into a thermally-stable triple helix? On a related point, do HeLa cells synthesise Hsp47, which is a collagen chaperone that binds TANGO1?

4) Figure 4: quantify the co-localisation (TANGO1/Sec16A).

5) Figure 5: the TANGO1 mutation is dominant over the wild type protein. Is over expression rather than the mutation of TANGO1 causing the collagen secretion defect?

6) The defects of the TANGO1 mutation are pleiotropic and therefore the syndrome should not be called collagenopathy. It introduces a wrong bias in the reader's mind.

7) Please perform a RIP-Chip to show reduced SR protein binding to the mutant Tango exon 8 encoding RNA.

8) Please test whether the mutation induces UPR.

9) Is it possible to stain for collagen deposition in tissues of affected individuals? If not, please discuss.

10) The authors suggest that hormone secretion is affected by the TANGO1 mutation. This should be shown or the text modified.

[Editors' note: further revisions were suggested prior to acceptance, as described below.]

Thank you for re-submitting your article "Biallelic TANGO1 mutations cause a novel syndromal disease due to hampered cellular collagen secretion" for consideration by *eLife*. Your article has been re-reviewed by three peer reviewers, including Reinhard Faessler as the Reviewing Editor and Reviewer #1, and the evaluation has been overseen by Suzanne Pfeffer as the Senior Editor. The following individual involved in review of your submission has agreed to reveal their identity: Karl E Kadler (Reviewer #3).

The reviewers have discussed the reviews with one another and the Reviewing Editor has drafted this decision to help you prepare a revised submission.

The authors report the complex disease phenotype of a patient carrying a homozygous TANGO1 gene mutations leading to the expression of a truncates and functionally disabled TANGO1 protein. The reviewers discussed the revisions and concluded that the authors still need to provide some corrections and explanations prior to acceptance of the manuscript for publication in *eLife*.

Essential revisions:

1) The complex disease phenotype (e.g. defects in dentinogenesis/dentinogenesis imperfecta, diabetes, hearing loss, etc.) due to the TANGO1 mutation is probably not only caused by the aberrant export and secretion of collagens. Dentinogenesis imperfecta for example, can be caused by collagen type I defects (osteogenesis imperfecta, often associated with hearing loss) or mutations in the DSPP gene coding for the dentin sialophosphoproteins. This example can be used in the Discussion to alert readers that TANGO1 does more than only controlling the secretion of collagens and that the complex phenotype may be due to the aberrant export of diverse proteins.

2) The Western blots do not show a convincing chance in collagen XII and IV levels, nor do the authors report the technical repeats and statistics with these antibodies. This information needs to be provided. It is also necessary to mention the source of the collagen XII and IV antibodies.

3) Discuss how does mutated TANGO1 or altered levels (?) of TANGO1 expression lead to decreased Col1 transcription or steady-state levels of Col1xx. It should also be reported which Col1 chain mRNAs, COL1A1 and/or COL1A2, have been analysed.

4) proa1(I) or a1(I) are the proteins; *COL1A1* and *COL1A2* are the genes. Please correct accordingly.

---

## [Author Response]

Essential revisions:1) The TANGO1 mutation profoundly affects collagen I secretion, although a regulation of collagen I secretion was not observed in Saito et al. What is the explanation of this discrepancy?

We appreciate this concern, but many others including our own data now show that TANGO1 is required for export of many collagens including Collagen 1. We do not know whether it was the serum, the cells, the growth conditions, the siRNA reagents, or the antibodies used to detect collagens that led us astray in our earlier descriptions of TANGO1. With this report, we correct our previous mistake and hope that we can move forward with the accepted role of TANGO1 in export of collagens in general.

We have explicitly stated this in the Discussion. It now reads, “In two previous descriptions of TANGO1function (Nogueira et al., 2014; Saito et al., 2009), we reported that it had no obvious role in collagen I export. However, it has been demonstrated subsequently that TANGO1 functions to export all soluble collagens tested thus far, including collagen I”.

2) Determine whether also the secretion of additional collagens, collagen II and VII are affected by the mutation, ideally in patient-derived fibroblasts.

We have included data showing the effect on the secretion of collagen IV and collagen XII in U2OS cells. We have no access to patient-derived tissue.

3) Why were HeLa cells used in Figure 3 when U2OS cells would have been a better cell choice? HeLa are not a good cell choice for studies of collagen. Do HeLa cells express prolyl 4-hydroxylase, which is essential for procollagen folding and assembly into a thermally-stable triple helix? On a related point, do HeLa cells synthesise Hsp47, which is a collagen chaperone that binds TANGO1?

We have used U2OS cells and modified Figure 4 accordingly and removed the data obtained from HeLa cells. In answer to the reviewer query, HeLa cells do express both P4HA1 and HSP47. Data from Itzhak et al. (*eLife*, 2016) show copy numbers of P4HA1 and HSP47 as 646,700 and 8,094,200 molecules per cell respectively.

4) Figure 4: quantify the co-localisation (TANGO1/Sec16A).

The colocalisation (or lack thereof) has been quantified and plotted in Figure 4 alongside the immunofluorescence images.

5) Figure 5: the TANGO1 mutation is dominant over the wild type protein. Is over expression rather than the mutation of TANGO1 causing the collagen secretion defect?

This point is already discussed:

"Our study provides evidence that aberrant expression of a truncated TANGO1 protein and/or reduced levels of fully functional TANGO1 protein, or likely a combination of both causes a novel syndrome due to disturbances in cellular protein secretion. […] This is consistent with a threshold model, where the disease only manifests when the ratio of truncated versus normal protein exceeds a critical level."

In addition, we have carried out an added experiment to demonstrate the specificity of the effect of the truncated TANGO1 on collagen secretion. We now show that overexpression of full-length TANGO1 in the same system shows no effect on the secretion of collagen 1. These data are included as Figure 5—figure supplement 3.

6) The defects of the TANGO1 mutation are pleiotropic and therefore the syndrome should not be called collagenopathy. It introduces a wrong bias in the reader's mind.

We have removed references to a collagenopathy when appropriate.

7) Please perform a RIP-Chip to show reduced SR protein binding to the mutant Tango exon 8 encoding RNA.

In our opinion the proposed RIP-Chip or RIP-Seq experiment to show binding of SR proteins to TANGO1 (pre)mRNA is not feasible. Prerequisite to this experiment would be the establishment of a stable mutation cell line or to use patient derived cells for the analyses. We performed a number of unsuccessful experiments to establish a mutation-specific TANGO1 cell line via CRISPR and/or homologous recombination. Moreover, we are unable to get access to patient derived cells. An alternative experiment would be using transiently transfected or stably overexpressing cells, which would have the disadvantage of harboring unnaturally high amounts of TANGO1 transcripts, subsequently resulting in unspecific binding to SR proteins and in misleading RIP-Seq results.

8) Please test whether the mutation induces UPR.

We include data (Figure 5—figure supplement 1) to show that the expression of the truncated TANGO1 does not induce UPR. We show that there is no change in XBP1 splicing under these conditions.

9) Is it possible to stain for collagen deposition in tissues of affected individuals? If not, please discuss.

Unfortunately, we are unable to obtain patient-derived cells.

10) The authors suggest that hormone secretion is affected by the TANGO1 mutation. This should be shown or the text modified.

We have removed the statement from the manuscript but retained the discussion of literature showing a role of cTAGE5 and TANGO1 in insulin homoeostasis.

[Editors' note: further revisions were suggested prior to acceptance, as described below.]Essential revisions:1) The complex disease phenotype (e.g. defects in dentinogenesis/dentinogenesis imperfecta, diabetes, hearing loss, etc.) due to the TANGO1 mutation is probably not only caused by the aberrant export and secretion of collagens. Dentinogenesis imperfecta for example, can be caused by collagen type I defects (osteogenesis imperfecta, often associated with hear loss) or mutations in the DSPP gene coding for the dentin sialophosphoproteins. This example can be used in the Discussion to alert readers that TANGO1 does more than only controlling the secretion of collagens and that the complex phenotype may be due to the aberrant export of diverse proteins.

We have added the following paragraph to the Discussion: “Type I collagens provide tensile strength to connective tissue and are abundant in bone, skin, dentin, cementum, tendons, and ligaments (Deshmukh et al., 2016). […] For example, mutations in the dentin sialophosphoprotein (DSPP) gene and, by extrapolation, TANGO1-associated defects in DSPP secretion, may cause DGI 1 (#605594) with or without hearing loss (Xiao et al., 2001) as well as DGI, Shields type II (#125490) and III (#125500), which may be phenotypic variation of the same entity rather than separate diseases (Kim et al., 2005).”

2) The Western blots do not show a convincing chance in collagen XII and IV levels, nor do the authors report the technical repeats and statistics with these antibodies. This information needs to be provided. It is also necessary to mention the source of the collagen XII and IV antibodies.

By quantifying the relative amount of collagen I present in the cells and in the media at each time point, a drastic reduction in the rate of collagen I secretion from the EX8-HA stable cell line compared to control cells was observed. Importantly, this effect was not due to a general reduction of protein secretion since the small cargo antitrypsin was produced and secreted at a comparable rate in the two cell populations. Under the same conditions, we tested secretion of collagen IV and collagen XII in EX8-HA stable line. 6 independent trials revealed a consistent trend of a net reduction in their secretion, but because of the low relative abundance of these collagens the magnitude of the effect varied considerably and cannot be statistically quantified (Figure 5—figure supplement 2). These secretion defects were specific to Ex8 expression as we observed no change in collagen I secretion when full-length TANGO1 was overexpressed in U2OS (Figure 5—figure supplement 3). Collectively, these results show that expression of the exon 8 skipped splice product even in the presence of full length TANGO1 affects collagen I homeostasis.

The source of antibodies is now stated in the Materials and methods.

3) Discuss how does mutated TANGO1 or altered levels (?) of TANGO1 expression lead to decreased Col1 transcription or steady-state levels of Col1xx. It should also be reported which Col1 chain mRNAs, Col1a1 and/or Col1a2, have been analysed.

We have noticed a change in the mRNA level of collagen α1 chain in U2OS cells expressing mutant TANGO1. Although, we have not tested the effects on the expression of other chains of collagen 1 and other collagens, it is conceivable that expression of this mutant causes accumulation of collagens in the ER, which then affect their further synthesis by a feedback mechanism. However, this appears not to involve the activation of UPR in U2OS cells. Whether this is also the situation in vivo in the individuals with TANGO1 mutation is unknown.

4) pro-α1(I) or α1(I) are the proteins; COL1A1 and COL1A2 are the genes. Please correct accordingly.

Done.